# Use of a Novel Real-Time PCR to Investigate Anthelmintic Efficacy Against *Haemonchus contortus* in Sheep and Goat Farms

**DOI:** 10.3390/vetsci12060569

**Published:** 2025-06-10

**Authors:** Anna Maurizio, Giorgia Dotto, Cinzia Tessarin, Paola Beraldo, Giovanni Franzo, Rudi Cassini

**Affiliations:** 1Department of Animal Medicine, Production and Health, University of Padova, Viale dell’Università, 16, 35020 Legnaro, Italy; anna.maurizio@phd.unipd.it (A.M.); giorgia.dotto@unipd.it (G.D.); cinzia.tessarin@unipd.it (C.T.); giovanni.franzo@unipd.it (G.F.); 2Department of Agricultural, Food, Environmental and Animal Sciences, University of Udine, Via Sondrio, 2/A, 33100 Udine, Italy; paola.beraldo@uniud.it

**Keywords:** molecular analysis, gastrointestinal nematodes, anthelmintic resistance, FECRT, sheep, goats, *Haemonchus contortus*

## Abstract

Gastrointestinal nematode infections are widespread in goat and sheep farming and their accurate diagnosis is rarely performed. Traditional methods have limitations, leading to a shift towards more sensitive molecular diagnostic techniques. This study presents the development of a novel real-time PCR method for diagnosing *Haemonchus* sp. infections in grazing ruminants, to estimate the relative abundance of this parasite in mixed infections. The method demonstrated good performance and it was then applied in Faecal Egg Count Reduction Test (FECRT) trials conducted on five sheep and five goat farms in northeastern Italy. The results indicate concerning levels of anthelmintic treatment ineffectiveness, with susceptibility confirmed in only three farms. This highlights the widespread misuse of anthelmintics and the associated risk of resistance development. Combining molecular tools with conventional parasitological techniques can enhance disease surveillance and provide more rapid responses for managing and controlling gastrointestinal nematode infections. The study highlighted an urgent need for training local farmers and veterinarians on best treatment practices, encouraging the adoption of the newly developed method.

## 1. Introduction

The accurate and timely diagnosis of parasitic infections is crucial for effective disease management in livestock, whose productivity and animal welfare can be severely affected by these infections. Gastrointestinal nematode (GIN) infections are a major concern in small ruminant farming, leading to substantial economic losses and health issues within flocks [1]. These infections are due to a huge variety of genera and species of parasites that can cause a range of symptoms and even result in death if not properly managed. Monitoring parasite infections and evaluating anthelmintic effectiveness depends on accurate diagnostic tools. However, the application of species- or genus-specific diagnosis remains limited. Traditional diagnostic methods for GIN, i.e., faecal egg counts (FECs), do not allow for species-specific identification, and larval culture, though widely used for this latter purpose, has significant drawbacks [2]. This technique is labour-intensive, time-consuming, and often lacks the specificity required for accurate species identification due to the overlapping morphological features of certain nematode species.

In response to these challenges, there has been a rising shift towards molecular diagnostic techniques, which offer enhanced sensitivity, specificity, and speed in detecting and quantifying parasitic DNA. Since the work of Von Samson-Himmelstjerna et al. [3], various real-time polymerase chain reaction (PCR) assays have been developed [4,5,6]. Additionally, other techniques, such as multiplexed real-time PCR [7,8,9,10], LAMP assays [11,12], digital droplet PCR (ddPCR) [13], and nemabiome sequencing [14] have been explored. However, these assays do not include a quantification of the overall strongyles as a reference point for the quantification of the specific GIN species. As a result, relative quantification can be complex and not straightforward, particularly when not all nematode species are accounted for in the assay. Only Elmahalawy et al. [13] addressed this issue using ddPCR, but this technique requires expensive equipment that is not accessible to most laboratories.

To address these limitations and to integrate molecular diagnostics into its routine laboratory practices, the Laboratory of Parasitology and Parasitic Diseases at the University of Padova’s Department of Animal Medicine, Production, and Health developed a novel real-time PCR assay. In northern Italy, the status of GIN infections in small ruminants has been poorly investigated, but previous studies have highlighted the widespread presence of *Haemonchus contortus* [15], with local farmers reporting significant losses due to this parasite (personal communication). In addition, despite limited data for the region, two recent reports have documented cases of AR [16,17], with *H. contortus* playing a key role in the phenomenom. *H. contortus* is a highly pathogenic, blood-feeding nematode that can produce thousands of eggs daily, leading to rapid pasture contamination and outbreaks of haemonchosis. Unlike many other GINs, *H. contortus* does not typically cause diarrhoea, making its presence less detectable through routine observation. The rapid emergence of anthelmintic resistance in *H. contortus* populations [18] further underscores the need for accurate diagnosis that can guide veterinary practitioners.

The primary aim of this study was to develop and validate a novel real-time PCR assay that allows for the specific relative quantification of *H. contortus* in faecal samples from small ruminants. Unlike previously published molecular techniques, this assay includes a quantification of overall strongyles, which provides a reference point to contextualise species-specific measurements. This approach was developed using pooled faecal samples to improve cost-efficiency, and it was designed on standard real-time PCR to ensure broader accessibility, avoiding the need for more advanced and costly equipment. The novel real-time PCR assay presented in this study was designed for *H. contortus*, but it has the potential to be expanded to other species. The assay was then applied to faecal samples collected from sheep and goat farms in northern Italy in conjunction with ongoing Faecal Egg Count Reduction Tests (FECRTs) to assess the response of parasites to anthelmintic treatment in the area.

## 2. Materials and Methods

### 2.1. Real-Time PCR Development

The protocol presented is based on the setup of two real-time PCRs, one generic for all strongyles (GEN real-time PCR) and one specific for *Haemonchus* sp. (HAEM real-time PCR).

#### 2.1.1. Primer and Probe Design

Primers/probe sets targeting all strongylids (GEN) and specific for *Haemonchus* sp. (HAEM) were designed on the basis of the alignment of the 18S-rRNA-ITS1-5.8S-ITS2 region downloaded from the NCBI GenBank database (https://www.ncbi.nlm.nih.gov/genbank/ accessed on 16 January 2023) and aligned using the Clustalw method (https://npsa.lyon.inserm.fr accessed on 16 January 2023) (Table 1). This region contains heterogeneous sequences, flanked by areas of high similarity among strongyles. PRIMER3 software version 4.0 (https://primer3.ut.ee/ accessed on 16 January 2023) was used to select the best sets of primers and probes, setting stringent criteria to minimise melting temperature delta, primer self-complementarity, and repeated region occurrence.

To ensure the specificity of the oligos designed, in silico PCR was performed for all primer/probe combinations by using BLASTn against the NCBI GenBank. The primer/probe sets were synthetized using Macrogen Europe (Amsterdam, The Netherlands) (www.macrogen-europe.com accessed on 16 January 2023) and were then stored at −20 °C.

#### 2.1.2. Specificity of the Method

Samples of adult nematodes and third stage larvae (L3) were used to test the capacity to include all genera for the GEN assay and only the *Haemonchus* genus for the HAEM assay. Adult parasites were recovered during a previous research project from the gastrointestinal tract of a roe deer and coming from selective hunting [19]. Parasites were identified according to existing morphological and morphometric keys, as detailed in Gibbons and Khalil [20]. Adult nematodes were cut in two parts, and half was preserved as non-extracted for future reference. L3 were isolated via the modified Baermann technique from sheep and goat faeces cultured with vermiculite for 7 days at 26 °C and were identified as described in Maurizio et al. [21]. Adult parasites and L3 were identified as *Trichostrongylus* sp., *Chabertia* sp., *Cooperia* sp., *Teladorsagia* sp. *Strongyloides* sp., *Oesophagostomum* sp., *Bunostomum* sp., and *Haemonchus* sp. and stored individually at −20 °C. For the isolation of genomic DNA, individual adult worms and L3 were thawed and processed with the Macherey-Nagel™ NucleoSpin™ Tissue Kit (standard protocol for human, animal tissue, and cultured cells). Briefly, the first step consisted of pre-lysis with 180 µL Buffer T1 and 25 µL Proteinase K solution, with overnight incubation at 56 °C. Samples were then lysated with Buffer B3 at 70 °C for 10 min and precipitated with 210 µL ethanol (96–100%). The next steps of binding, washing, and DNA elution were performed according to the manufacturer’s instructions. The eluted DNA was then stored at −20 °C until further analysis.

The concentration and purity of extracted DNA were measured using Invitrogen™ Qubit™ 4 Fluorometer (Fisher Scientific, Waltham, MA , USA). Genus identification was confirmed by end-point PCR, as described by Bott et al. [2] and Wang et al. [22]. The amplicons were finally sequenced using Sanger technology (Macrogen, Madrid, Spain) and compared with those already published in GenBank.

Both sets of primers/probes (GEN and HAEM) were tested via real-time PCR for specificity against DNA extracted from each of the above-mentioned genera. A GIN-negative ovine faecal sample was included as a negative control.

#### 2.1.3. Sensitivity of the Real-Time PCR Method

Nine thousand purified *H. contortus* eggs in 1 mL of PBS solution originating from a monospecific experimental infection of a lamb were kindly provided by the Czech University of Life Sciences. The concentration of eggs per 100 μL of the solution was estimated by filling 10 McMaster slides, counting the eggs per slide, calculating the average of the obtained counts, and adjusting the result to the dilution factor. The extraction of genomic DNA was carried out on the original sample as described in Section 2.1.2, and the eluted DNA was stored at −20 °C until further analysis.

The sensitivity, efficiency, and the determination coefficient (R^2^) of the assay were tested using 10-fold serial dilution of the obtained DNA. Sensitivity, efficiency, and error were determined by calculating the limit of detection (LOD), defined as the lowest concentration where the positive samples were detected in 50% of the replicates.

To include the extraction efficiency in the quantification, allowing for a proper comparison with field samples, a 10-fold dilution (9000 to 0.9 eggs/μL) of the purified eggs was performed, and subsequently, DNA extraction was carried out as described in Section 2.1.2 for each dilution. Two replicates for each DNA sample were then tested with the GEN real-time PCR protocol and two other replicates for each DNA sample were tested with the HAEM real-time PCR protocol, and the mean cycle time (Ct) values were calculated. To assess the assay repeatability, all five dilutions were assessed over three different days and the coefficient of variation (CV) was calculated for each egg dilution and tested with both assays across multiple runs. In order to assess if significant differences among runs occurred, a GLM model was fitted, including the dilution, run, and specific test as explanatory variables. The significance level for the statistical tests was designated at *p* < 0.05.

A complete standard curve was added to each experiment to calculate the run-specific efficiency and allow for the precise quantification of unknown sample.

#### 2.1.4. Real-Time PCR Conditions

Two replicate real-time PCRs were carried out with each of the primer/probe sets (GEN and HAEM). The PCRs were carried out in a LightCycler LC96 (Roche, Basel, Switzerland) in a total volume of 20 µL containing 1X QuantiNova Probe RT-PCR Mastermix (Qiagen, Hilden, Germany), 0.8 µM of each primer, 0.2 µM of specific probe, and 2 µL of DNA. Molecular-biology-grade water was added up to the final volume. The thermal cycle consisted of 1 cycle at 95 °C for 5 min; 40 cycles of 95 °C for 5 s and 60 °C for 30 s. The data were analysed using LC96 software version 1.1.

### 2.2. Faecal Egg Count Reduction Tests

#### 2.2.1. Study Design

The use of the new real-time PCR was then tested in naturally infected sheep and goats during FECRT trials in small ruminant farms of northeastern Italy. All trials were carried out between March 2023 and January 2024 in compliance with the WAAVP guidelines [23]. Fifteen animals were selected for each farm and a faecal sample was collected from them at day 0 (D0), when they also received the anthelmintic treatment, and then 10 to 14 days later (D14), depending on the drug used [23]. Faecal samples were collected from the rectum and individually identified using the ear tag codes. Sampling and treatment were carried out by trained veterinary practitioners. Each veterinary practitioner was provided with a detailed protocol and trained by staff from the University of Padova. The dosage was calculated on the heaviest animal of the group. Weight estimation was based on visual assessment. The accuracy of this approach was evaluated and confirmed by weighing a few random animals during the training phase using a dynamometer and a weighing harness. Veterinary practitioners then proceeded autonomously, and the shipment of samples to the Parasitology Laboratory of the Department of Animal Medicine, Production and Health of the University of Padova was organised thanks to the support of the Istituto Zooprofilattico Sperimentale delle Venezie. All samples were kept under cold chain and either processed within a maximum of 48 h after collection or vacuum-packed and analysed within two weeks. Anthelmintics were independently selected by the farm veterinarians, according to the treatment plan in use at each specific farm, as follows:Avermectines (AVMs):
Tolomec^®^ (ivermectin)—solution for subcutaneous injection, FATRO S.p.A., Ozzano dell’Emilia, Italy (farms S7, S8 and S11);Ecomectin^®^ (ivermectin)—solution for subcutaneous injection, ECO Animal Health Europe Limited, London, United Kingdom (farm S9);Oramec^®^ (ivermectin)—oral suspension, Boehringer Ingelheim Animal Health Italia S.p.A., Milano, Italy (farm G7);Eprinex Multi^®^ (eprinomectin)—pour-on, Boehringer Ingelheim Animal Health Italia S.p.A., Milano, Italy (farms G8 and G9).
Benzimidazoles (BZs):
Panacur 2.5%^®^ (fenbendazole)—oral suspension, MSD Italia, Roma, Italy (farms G10 and S10).Imidazothiazoles (LEVs)
Toloxan^®^ (levamisole)—oral suspension, FATRO S.p.A., Ozzano dell’Emilia, Italy (farm G11).


Treatments were performed according to the manufacturer’s instructions, and double the ovine dosage was used in goats. Only farms in which at D0 at least 200 eggs were counted in total (prior to transformation to EPG) among the 15 animals, were included. For each farm, 5 g from each individual sample, when available, was pooled together. The pooled sample was then mixed, thoroughly homogenised, and split in two (pool A and pool B). Pooled samples A and B were processed separately for all the subsequent steps. When it was not possible to sample an animal at D14, the animal was excluded from the entire trial. In all trials, at least 10 animals were included.

#### 2.2.2. Laboratory Analysis

Both individual and pooled samples were analysed with the Mini-FLOTAC technique (detection limit: 5 EPG) [24], using a sucrose–sodium nitrate flotation solution (specific gravity = 1.300). Pooled samples were further processed to obtain purified eggs for molecular analysis.

Purified strongylid eggs were obtained from pooled samples through sedimentation–flotation. Briefly, 5 g of each sample was mixed with H_2_O and entirely filtered through three gauzes and a sieve in a Falcon tube, which was subsequently centrifuged at 1900 rpm for 4’. The supernatant was then removed with a pipette and the sediment resuspended with a sucrose–sodium nitrate solution (specific gravity 1.300). The Falcon tube was centrifuged again with the same settings and filled up with the sucrose–sodium nitrate solution. A slide was then applied on top [25], in contact with the solution, left there for 5’, and then carefully washed with H_2_O in a new Falcon tube. This procedure was repeated three times. Finally, the second Falcon tube was centrifuged at 1900 rpm for 5’ and the supernatant eliminated with a pipette. The sediment, containing purified eggs, was then stored at −20 °C until DNA extraction, which was performed as described in Section 2.1.2, except for the reagents’ amount, which was increased to 400 µL for Buffer T1, 50 µL for Proteinase K, and 300 µL for Buffer B3 and ethanol.

Samples were then processed in duplicate in each real-time PCR assay, as described in Section 2.1.4. A ten-fold dilution of the purified eggs was included in each run to quantify GIN/*Haemonchus* sp. eggs of positive samples. The Bland–Altman plot was used to graphically visualise the agreement between the two replicates (i.e., the two pools of each sample), as a proxy for the repeatability of the method.

#### 2.2.3. Data Analysis

FECR (Faecal Egg Count Reduction) and 90% confidence intervals (90% CI) were calculated, for both general FECRTs and genus-specific FECRTs, according to one of the methods [26] indicated in the recently revised WAAVP guidelines [23]. Anthelmintic treatment efficacy was then interpreted according to the classification outlined by Denwood et al. [27], and subsequently integrated into the WAAVP guidelines. The minimum efficacy target and expected efficacy, respectively, were maintained at 90% and 95% [28] and, as such, the classification was as follows:Resistant (R) when the upper limit of the 90% CI (CIU) < 95%;Low resistance (LR, a sub-category of the previous) when CIU < 95% and the lower limit of the 90% CI (CIL) ≥ 90%;Inconclusive (INC) when CIU ≥ 95% and CIL < 90%;Susceptible (S) when CIU ≥ 95% and CIL ≥ 90%.

The relative proportion of *Haemonchus* sp. eggs was obtained by calculating the ratio between the number of *Haemonchus* sp. and strongyles eggs estimated through absolute quantification performed using the HAEM and GEN real-time PCRs, respectively. For genus-specific FECRTs, the estimated proportion of *Haemonchus* sp. eggs was then converted into the absolute number of eggs, considering the total number of eggs counted (prior transformation to EPG) among the 15 animals of the trial as a reference amount, as described in Maurizio et al. [21]. Genus-specific FECRTs and associated efficacy were calculated and interpreted as described above for the overall FECRT.

## 3. Results

### 3.1. Real-Time PCR Development

The sequencing of amplicons from the end-point PCRs confirmed the identification of adult strongylids as *Haemonchus* sp., *Teladorsagia* sp., *Trichostrongylus* sp., *Oesophagostomum* sp., *Chabertia* sp., *Cooperia* sp., *Strongyloides* sp., and *Bunostomum* sp. The GEN primer set was evaluated for its capacity to amplify DNA from all the above-mentioned parasites and *Haemonchus* sp. eggs, while the HAEM primer set was assessed for its specificity towards the *Haemonchus* genus. Amplification was achieved for all genera with the GEN primers, while only for adult parasites and purified eggs of *Haemonchus* sp. with the HAEM primers. The sensitivity and the efficiency of the real-time PCRs were determined by testing serial dilutions of genomic DNA from the most concentrated purified *H. contortus* eggs (9000 eggs). Amplification was achieved with as little as 0.9 eggs (LOD).

The GEN and HAEM real-time PCR assays demonstrated high efficiency, between 99 and 100%, with R^2^ values of 0.99 and 0.97, respectively. Standard curves were defined for the real-time PCR for overall strongylids and for *Haemonchus* sp. using a 10-fold serial dilution of purified *Haemonchus* sp. egg solution (from 9000 to 0.9 eggs, where each dilution was extracted separately). By including the extraction in the validation process, the standard curves determined for each assay (GEN and HAEM) were still proven repeatable on different days and involving different operators. However, the assays showed a lower efficiency, i.e., both 82%, with R^2^ values > 0.99. The assay repeatability was proven by the low CV, always lower than 5% regardless of the considered assay or dilution. Moreover, while a significant effect of dilution and assay was proven, no significant effect of the run was detected by the GLM model.

### 3.2. Real-Time PCR on Field Samples

Pooled faecal samples (n = 19) from goats and sheep naturally infected with GIN were analysed, with the results presented in Table 2. Overall, 38 real-time PCR analyses (two replicates for each pool) were conducted for both GEN and HAEM assays, but the data in Table 2 are presented in an aggregated form, showing the means between the two replicates (raw data are available in Appendix A). The findings indicate the presence of mixed infections in 100% of the samples. Using the Ct values and standard curves, the total number of strongylids and *H. contortus* eggs in each sample was estimated, allowing for the determination of the relative abundance of *H. contortus*.

Each composite sample underwent a completely separate process in duplicate (replicate 1 and 2), from egg isolation to DNA extraction and real-time PCR. The disaggregate data are reported in Appendix A. The relative abundance of *H. contortus* between the two replicates of the same trial was very similar for all trials. A visual representation of the agreement between the two replicates is shown in Figure 1, thanks to a Bland–Altman plot.

The estimated number of *H. contortus* eggs from the HAEM real-time PCR was consistently lower than the estimates from the GEN real-time PCR, except in two cases (trial G9, D0 and trial S10, D14). The percentage of *H. contortus* obviously cannot exceed 100% of the overall strongylids, and these results are probably due to variance in Ct interpolation. In most trials, the difference between the two replicates is minimal, apart from one outlier, which is due to the very low estimated numbers of eggs for both GEN and HAEM (trial S10, D14; see Appendix A).

### 3.3. Faecal Egg Count Reduction Tests

FECRTs were conducted on five goat and five sheep farms across the regions of Friuli—Venezia Giulia (Udine and Pordenone Provinces), Veneto (Verona, Vicenza, and Belluno Provinces), and Trentino—Alto Adige (Trento Province) in northeastern Italy. The treatment was effective in two out of five sheep trials and in one out of five goat trials (Table 3). Each of the three different anthelmintic classes tested showed effectiveness in one trial. The lowest effectiveness in goats was observed with pour-on eprinomectin, resulting in FECRT values of 24.7% and 46.7% in trials G8 and G9, respectively. Additionally, one of the three commercial ivermectin products demonstrated minimal efficacy in sheep, with FECRT values of −4.9% and 45.2% in trials S7 and S8, respectively.

For each trial, the FECRT for *H. contortus* was determined by correlating the real-time PCR results with copromicroscopic data. The *Haemonchus*-specific FECRT results revealed varying responses to the anthelmintic treatments across the trials. In trials G7, G8, S9, and S10, the presence of *H. contortus* decreased proportionally to the overall strongyles. However, in trials G10, S7, S8, and S11, *H. contortus* showed a lower response to treatment compared to the overall strongyles, indicating a potential resistance. Due to operational constraints, molecular analysis of the D14 samples could not be performed for trial G11.

## 4. Discussion

In this study, we developed a novel real-time PCR method for the diagnosis of *Haemonchus* sp. infection in grazing ruminants. This was achieved using two specifically designed combinations of primer/probe sets, which target the ITS1-5.8S-ITS2 region. One of these molecular targets is shared (GEN) by all strongylids, while the other (HAEM) is specific for the *Haemonchus* genus, enabling the calculation of the relative abundance of this parasite in mixed infections. *Haemonchus contortus* is the only species of this genus in sheep and goats, and represents a very significant parasitic nematode, causing severe anaemia, weight loss, and potentially fatal outcomes in these species, alongside substantial economic losses. Additionally, the widespread development of anthelmintic resistance among *H. contortus* populations exacerbates the challenge of controlling this parasite [18]. In southern Europe, the warm and humid climate fosters its proliferation, making it a critical focus for research and integrated parasite management strategies in southern Europe [29].

Coproculture is traditionally used to differentiate among GIN species in a mixed infection, but it has several intrinsic limitations, related to the time required, the different environmental conditions necessary for the larval development of individual nematode species [30], and the inability to unequivocally differentiate particular genera/species [31]. Although lectin-binding assays with PNA have been developed and refined to specifically stain and differentiate *H. contortus* eggs [32], their adoption in diagnostic laboratories has also been limited. More recently, specific lectins have also been identified for other nematode species such as *Teladorsagia circumcincta*, *Oesophagostomum* sp., *Nematodirus* sp., *Trichostrongylus* sp., and *Chabertia ovina* [32,33]; however, the requirement for advanced microscopy equipment (microscope with fluorescent light) and the labour-intensive nature of the procedure have hindered its widespread use [34]. Molecular tests offer significant advantages in terms of sensitivity, accuracy, speed, and potentially even cost, which make them ideal for use in diagnosing resistance in field settings. Molecular tests are currently useful tools in research settings, but their use in field diagnostics is almost absent. The purpose of our work was to develop a real-time PCR to employ as a diagnostic technique in our laboratory, as part of the diagnostic service for local farms. It is currently developed only for *Haemonchus* sp., as it is the main genus of interest and the diagnostic purpose forces a containment of costs, but it has the potential to be extended to other genera/species.

Our method demonstrated optimal efficiency and determination coefficients when applied to serial dilutions of the DNA extract. However, when the solution containing purified eggs was diluted prior to DNA extraction, and extractions were performed separately on each dilution, a reduction in efficiency was observed, while the correlation coefficient values, reflecting the reliability of the results, remained optimal. This reduction in efficiency may be attributed to factors associated with the extraction process. Despite this, both real-time PCR assays (GEN and HAEM) showed comparable efficiency and correlation, and considering the objective of achieving relative rather than absolute quantification (which is more susceptible to lower efficiency), the results can be considered reliable. Repeatability when processing the two pools obtained from the same animals (pool 1 and pool 2) separately was also very high. Future efforts should aim to improve the extraction efficiency to enhance the assay sensitivity and accuracy. The purification step prior to extraction is time-consuming, but it allowed us to obtain better results, especially in reducing egg loss and possible faecal inhibitors.

Some researchers suggest that the molecular amplification of faecal eggs may be affected by variations in the DNA content due to the developmental stage of the eggs and differences in the target gene copy numbers across species. However, these factors are generally not considered to be major sources of error [35]. Instead, much of the variability in quantification between samples with the same egg counts appears to be primarily attributed to the DNA extraction method. While the time since egg embryonation and the presence of competing DNA from closely related nematode species can influence quantification, significant changes in the DNA content that affect results are largely restricted to the first 6–7 h after embryonation [36].

This real-time PCR method was applied in FECRT trials conducted on five sheep and five goat farms in northeastern Italy. The results, combined with previous data from the same area [17] and from neighbouring regions [16,37], indicate concerning levels of treatment ineffectiveness. This is particularly alarming as it represents a strong driver for the development of AR, if not already present, and could pose a serious threat to the sustainability of the peculiar farming systems of the study area (i.e., transhumant farming, now recognised as UNESCO Intangible Cultural Heritage, and alpine farming). These systems are based on the extensive grazing of animals, which are continuously exposed to parasite infection and rely on pharmacological treatments to control heavy infections. Treatment failure is a strong indicator of AR; however, treatment can fail for reasons other than heritable AR in the parasite population, and these reasons should always be accounted for [38].

In the previous study [17], *H. contortus* consistently showed the worst responses to treatment, supporting the suspicion of resistance. In the current study, *H. contortus* showed a significantly lower response to treatment compared to the overall strongyle population in four out of ten trials (G10, S7, S8, and S11). In trials S7 and S8, *H. contortus* egg counts at D14 were even higher than those at D0. The cause of this increase remains unclear. However, despite *H. contortus* showing a markedly lower FECR percentage compared to the overall strongyles, it represented only a minority of the eggs detected at D14, so other genera were likely involved in the treatment failure. In trial S11, the FECR result for *H. contortus* was not considered meaningful due to the very low numbers detected. Except for farm G10, where the overall FECR of 79.3% was primarily driven by a lack of response to treatment in *H. contortus* (FECR = 4%), the percentage of reduction in the number of *H. contortus* eggs in the remaining trials was similar to the reduction in the overall strongyle egg counts. Although it is challenging to fully account for the imprecision of the data obtained through molecular assays (as it is for data obtained from coprocultures), these findings strongly suggest widespread treatment failure, rather than AR, which typically affects only one or two species in a mixed GIN population. This interpretation is also supported the observation that susceptibility was confirmed in only three farms. One of these, G11, was treated with levamisole for the first time, following the veterinarian’s suspicion of AR to benzimidazoles (BZs) and macrocyclic lactones (MLs), which had traditionally been used annually. Another, S10, had animals treated with BZs for the first time, after years of using praziquantel, of whose ineffectiveness the farmer was unaware against GIN. These results highlight not only the widespread misuse of anthelmintics and the associated risk of AR development to which the entire farming sector is exposed, but also the value of a genus-specific approach in interpreting treatment outcomes.

## 5. Conclusions

Combining molecular tools with conventional parasitological techniques, such as FEC and FECRT, can enhance disease surveillance and provide more rapid responses for managing and controlling GIN infections. These tools overcome the limitations of traditional methods (i.e., coproculture) and enable the more precise monitoring of species of concern, such as *H. contortus*. This not only supports epidemiological studies, but also holds promise for future use in field diagnostics. Further research is required to fully validate our method and encourage its adoption by local farmers and veterinarians. More specifically, the assay should be tested with a larger sample size and across different geographical regions, and the real-time PCR result compared with other diagnostic methods/techniques, also in terms of cost-effectiveness. Furthermore, our findings reveal the improper use of anthelmintics, underscoring the need to monitor changes in *H. contortus* populations and AR over time and to urgently improve training on best treatment practices. Finally, modern platforms like social media can play a valuable role in spreading accurate information about parasite diagnosis and AR beyond the scientific community [39], fostering transparency and public engagement through accessible, widely used channels.

## Figures and Tables

**Figure 1 vetsci-12-00569-f001:**
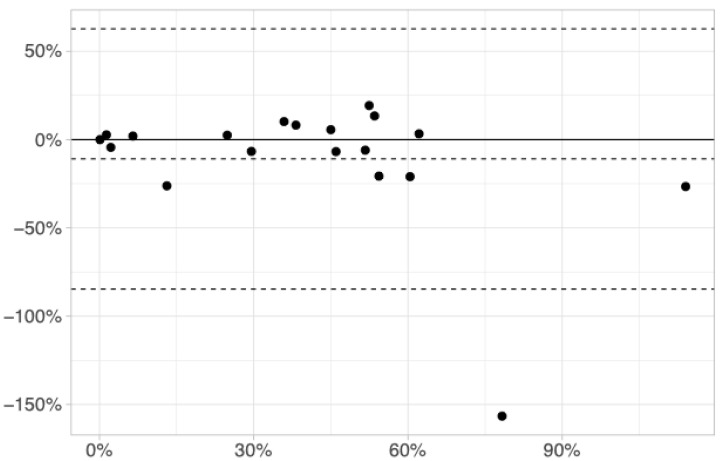
Bland–Altman plot of differences (*y*-axis) against means (*x*-axis) for the estimated percentage of *H. contortus* (HAEM) out of the overall strongylids (GEN), between the two replicates of each pooled sample (n = 19). The upper and lower dotted lines represent 95% confidence limits of the bias (central dotted line).

**Table 1 vetsci-12-00569-t001:** Sequences of selected primers and probes for overall strongylids (GEN) and specific for *Haemonchus* sp. (HAEM) real-time PCR assays.

Primer/Probe	Sequence (5′-3′)	Length of Amplicon
GEN		140 bp
Forward	5′-ATGGATCGGTTCGATTCGCGT-3′	
Reverse	5′-ACAACCCTGAACCAGACGTG-3′	
Probe	5′-(FAM) CGCATAGCGCCGTTGGGTTT (BHQ1)-3′	
HAEM		194 bp
Forward	5′-GGGCTAATTTCAACATTGTTGGT-3′	
Reverse	5′-ACATCGTCGCTATACATGTCAC-3′	
Probe	5′-(FAM) ACATGCAACGTGATGTTATGA (BHQ1)-3′	

**Table 2 vetsci-12-00569-t002:** Results of the molecular testing of pooled faecal samples with GEN real-time protocol for overall strongylids and HAEM real-time protocol for *Haemonchus* sp., for each trial at D0 and D14. Data are presented as mean between results obtained for the two replicates of each pool. Estimated quantity = number of eggs in the sediment, estimated by interpolating the cycle threshold with the standard curve.

Trial ID	Time	GEN Estimated Quantity (Mean)	HAEM Estimated Quantity (Mean)	Percentage *H. contortus* (%)
G7	D0	3132.0	1660.5	53.0
	D14	1177.5	728.5	61.9
G8	D0	450.5	206.5	45.8
	D14	142.0	64.5	45.4
G9	D0	1809.5	2171.0	120.0 *
	D14	1659.0	897.5	54.1
G10	D0	415.5	50.5	12.2
	D14	207.0	126.0	60.9
G11	D0	9.0	0.2	2.2
S7	D0	322.5	79.0	24.5
	D14	44.5	17.5	39.3
S8	D0	2905.5	197.0	6.8
	D14	2876.0	1017.0	35.4
S9	D0	289.5	86.5	29.9
	D14	195.0	100.0	51.3
S10	D0	503.5	261.0	51.8
	D14	1.5	2.5	161.3 *
S11	D0	402.0	0.4	0.1
	D14	7.5	0.2	2.0

* These percentages should be interpreted as equal to 100%.

**Table 3 vetsci-12-00569-t003:** FECR and 90% confidence intervals in trials of the study, calculated for overall strongylids and for *H. contortus.* FEC at D0 and D14 refer to total counted eggs among the 10–15 animals. ^a^ Oramec^®^, Boerhinger Ingelheim; ^b^ Eprinex Multi^®^, Boerhinger Ingelheim; ^c^ Panacur 2.5%^®^, MSD; ^d^ Toloxan^®^, Fatro; ^e^ Tolomec^®^, Fatro; and ^f^ Ecomectin^®^, ECO Animal Health Europe Limited. BZs = benzimidazoles; MLs = macrocyclic lactones; ITs = imidazothiazoles; Admin. route = administration route; SC = subcutaneous injection; PO = peroral; S = susceptible; R = resistant; n.a. = not available; and n.c. = not calculable.

	ID	Drug	Class	Admin. Route		FEC		FECR (%)	90% CI	Efficacy
					Genus	D0	D14			
Goat	G7	Ivermectin ^a^	ML	PO	Overall strongylids	5131	610	88.1	87.3–88.8	R
					*H* *. contortus*	2744	379	86.2	85.1–87.2	R
	G8	Eprinomectin ^b^	ML	Pour-on	Overall strongylids	741	539	27.3	24.7–30.0	R
					*H* *. contortus*	341	242	28.9	25.0–33.1	R
	G9	Eprinomectin ^b^	ML	Pour-on	Overall strongylids	8027	4205	47.6	46.7–48.5	R
					*H* *. contortus*	8027	2285	71.5	70.7–72.4	R
	G10	Fenbendazole ^c^	BZ	PO	Overall strongylids	1710	354	79.3	77.6–80.9	R
					*H* *. contortus*	223	213	4.4	2.7–7.3	R
	G11	Levamisole ^d^	IT	PO	Overall strongylids	2818	42	98.5	98.1–98.8	S
					*H* *. contortus*	62	n.a.	n.c.	n.c.	n.a.
Sheep	S7	Ivermectin ^e^	ML	SC	Overall strongylids	2389	2506	−4.9	n.c.	R
					*H* *. contortus*	594	957	−61.2	n.c.	R
	S8	Ivermectin ^e^	ML	SC	Overall strongylids	8705	4773	45.2	44.3–46.0	R
					*H* *. contortus*	566	1712	−202.5	n.c.	R
	S9	Ivermectin ^f^	ML	SC	Overall strongylids	235	42	82.1	77.6–85.8	R
					*Haemonchus*	69	22	68.3	58.5–76.5	R
	S10	Fenbendazole ^c^	BZ	PO	Overall strongylids	2627	0	100	99.9–100	S
					*H* *. contortus*	1359	0	100	99.8–100	S
	S11	Ivermectin ^e^	ML	SC	Overall strongylids	2827	55	98.1	97.6–98.4	S
					*H* *. contortus*	2.3	0.7	68.0	22.1–91.6	R

## Data Availability

The data supporting the reported results can be requested from the corresponding author.

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
