# Peer review of "Use of a Novel Real-Time PCR to Investigate Anthelmintic Efficacy Against Haemonchus contortus in Sheep and Goat Farms"

_vetsci, 2025, doi:10.3390/vetsci12060569_

Round 1
Reviewer 1 Report
Comments and Suggestions for Authors
Dear Authors,
By limiting the manuscript title to a specific region of a country, the authors create the impression that the new technique would only be valid in Northeastern Italy. I recommend removing the regional specification, as the title and objective of the study focus on standardizing a new diagnostic technique using real-time PCR.
The Simple Summary and Abstract briefly contextualize gastrointestinal nematode infections and the limitations of traditional diagnostics but lack explicit references to foundational studies (e.g., resistance mechanisms cited in DOI:10.1016/bs.apar.2016.02.012 or regional epidemiology from DOI:10.1590/S1984-29612021048).
Avoid repeating words already present in the text in the Keywords section. Please modify them accordingly.
Lines 58 to 61 contain 12 references, which is excessive from a scientific writing perspective. Try to reduce the number of references, keeping only the most relevant ones. The remaining references can be incorporated into the Discussion section.
Have the authors provided a sufficiently detailed description of the new technique (including reagents, equipment, volumes, and concentrations) to ensure it can be replicated and validated in other regions worldwide?
In line 155, the authors state that "Fifteen animals were selected for each farm, and a fecal sample was collected." However, the Institutional Review Board Statement does not mention an approved animal study protocol from the National Council for Control of Animal Experimentation (CONCEA) or approval from an Ethics Committee on Animal Use. Additionally, there is no mention of farmers' informed consent for the use of data in the publication. These documents are essential for submitting and approving an article involving biological samples.
Let’s be critical and didactic: Are three tables truly necessary? Could the main data be consolidated into a single table, with the remaining information described in the Results section? This would improve readability and comprehension. Overly complex tables with excessive data can be difficult to interpret, particularly for readers without a statistical background. An article should be easy to read and interpret to facilitate citation, replication, and application by the global scientific community.
The discussion lacks a crucial element: the financial feasibility of the newly standardized technique. There is little value in developing a highly sensitive and specific test if rural producers cannot afford it. The discussion should include a cost analysis comparing different diagnostic methods, performance times, and positive and negative predictive values.
A Conclusion section is missing. If the authors had followed the Veterinary Sciences journal template, they would not have made this fundamental omission.
Best regards
Author Response
Comment 1: Dear Authors, By limiting the manuscript title to a specific region of a country, the authors create the impression that the new technique would only be valid in Northeastern Italy. I recommend removing the regional specification, as the title and objective of the study focus on standardizing a new diagnostic technique using real-time PCR.
Reply: We agree with the reviewer and modified the title accordingly.
Comment 2: The Simple Summary and Abstract briefly contextualize gastrointestinal nematode infections and the limitations of traditional diagnostics but lack explicit references to foundational studies (e.g., resistance mechanisms cited in DOI:10.1016/bs.apar.2016.02.012 or regional epidemiology from DOI:10.1590/S1984-29612021048).
Reply: References are typically avoided in the Simple Summary and Abstract sections of research papers, except when essential. We believe our decision to omit references in these sections is consistent with standard publishing practices.
Comment 3: Avoid repeating words already present in the text in the Keywords section. Please modify them accordingly.
Reply: We modified the Keywords as per reviewer’s suggestion.
Comment 4: Lines 58 to 61 contain 12 references, which is excessive from a scientific writing perspective. Try to reduce the number of references, keeping only the most relevant ones. The remaining references can be incorporated into the Discussion section.
Reply: We appreciate the reviewer’s comment. However, we believe that the references included in the Introduction are important for providing a clearer understanding of the techniques currently available in the literature. As the purpose of the Introduction is to clarify context and background for the study, we consider these references useful as presented now.
Comment 5: Have the authors provided a sufficiently detailed description of the new technique (including reagents, equipment, volumes, and concentrations) to ensure it can be replicated and validated in other regions worldwide?
Reply: We further detailed the description of the new technique as per reviewer’s suggestion.
Comment 6: In line 155, the authors state that "Fifteen animals were selected for each farm, and a fecal sample was collected." However, the Institutional Review Board Statement does not mention an approved animal study protocol from the National Council for Control of Animal Experimentation (CONCEA) or approval from an Ethics Committee on Animal Use. Additionally, there is no mention of farmers' informed consent for the use of data in the publication. These documents are essential for submitting and approving an article involving biological samples.
Reply: Ethical review and approval were waived for this study, due to adoption of routine diagnostic sampling procedures. No experimental interventions or deviations from the standard clinical practice were involved. Ethical guidelines published by the International Council for Laboratory Animal Science (ICLAS) were followed when samples were collected on farms. Informed consent was verbally obtained from all owners. These statements are now reported in the relevant sections at the end of the manuscript, and we are sorry for having overlooked this aspect.
Comment 7: Let’s be critical and didactic: Are three tables truly necessary? Could the main data be consolidated into a single table, with the remaining information described in the Results section? This would improve readability and comprehension. Overly complex tables with excessive data can be difficult to interpret, particularly for readers without a statistical background. An article should be easy to read and interpret to facilitate citation, replication, and application by the global scientific community.
Reply: The three tables included in the manuscript address distinct aspects of the study: (1) sequences of primers and probes, (2) results of the molecular assay, and (3) results of the FECRT. While we acknowledge that Table 1 could technically be incorporated into the text, we believe that presenting the information in table form enhances clarity and facilitates quicker reference for readers. Regarding Tables 2 and 3, merging them into a single table would result in an overly complex and extensive format, potentially compromising readability. Similarly, integrating their content into the main text would significantly reduce the amount of detail presented, which we feel to be not functional to the purpose of the article. However, we agree with the reviewer that Table 2 could be simplified, and we revised it accordingly (keeping the availability of all data in a new supplementary material file: Table S1). The whole paragraph 3.2 has been revised to be consistent between the Table and the text, and a new Figure has been introduced to show graphically the agreement between the two replicates of each pool. The terms replicate 1 and replicate 2 has been used instead of pool 1 and pool 2, to avoid confusion.
Comment 8: The discussion lacks a crucial element: the financial feasibility of the newly standardized technique. There is little value in developing a highly sensitive and specific test if rural producers cannot afford it. The discussion should include a cost analysis comparing different diagnostic methods, performance times, and positive and negative predictive values.
Reply: The new technique was developed with the aim of minimizing costs (e.g., through the use of pooled samples and by targeting a single genus, Haemonchus contortus) and to integrate this technique in the routine laboratory diagnostic service. In our laboratory, we have observed cost advantages compared to traditional coproculture, which has a cost of 25 € for each sample Additionally, we have emphasized throughout the manuscript the reduced processing time associated with this technique compared to coprocultures. However, we have not conducted a detailed cost analysis, as we believe such an evaluation, based solely on data from our laboratory, would not be relevant to the wider audience. A comprehensive cost analysis involving multiple laboratories would indeed be valuable, but it was beyond the main scope of this study and would require a more extensive research effort. This aspect is now briefly addressed in the Conclusions section.
Comment 9: A Conclusion section is missing. If the authors had followed the Veterinary Sciences journal template, they would not have made this fundamental omission.
Reply: The Instructions for Authors of Veterinary Sciences state about Conclusions that “This section is not mandatory but can be added to the manuscript if the discussion is unusually long or complex”, and we opted for this solution in the original manuscript. However, the Conclusions section has been added in the revised version, also to include further suggestions provided by both Reviewers.
Reviewer 2 Report
Comments and Suggestions for Authors
The paper written by Maurizio et al. presents a research in the development and use of a novel real-time PCR method for the diagnosis and related quantification of Haemonchus sp., a significant parasitic nematode in graze ruminants. The authors highlight the importance of accurate and timely diagnosis of gastrointestinal nematode infections in livestock, emphasizing the limitations of traditional diagnostics tools and the increasing need for more sensitive and specific molecular techniques.
I found the subject interesting of this research to be fascinating and on time, and I read the manuscript with considerable interests. The paper addresses a critical need in veterinary parasitology, and the development of a method to quantify Haemonchus sp. relative to other strongylids is a valuable contribution. The paper aligns well with the scope of the journal, and the potential applications of this methodology in improving our understanding of anthelmintic resistance and parasite management are significant.
anyway, the study has many strongs, I think that the manuscript, in its present form, has several shortcomings that need to be addressed to enhance its clear, rigor, and general impact. My major comments to the authors, listed below, focus on aspects where I believe revisions are needed to support the conclusions of the research and to ensure the work meets the elevate standards of the journal.
Major Comments to the Author
Introduction
- The introduction give a well general overview of the problem of GIN infections and the limitations of traditional diagnostic techniques. However, it could benefit from a clearer statement of the specific aims of the study and the novelty of the developed PCR assay. Consider explicitly stating how this assay improves upon existing molecular diagnostic techniques, beyond just the inclusion of a strongyle quantification.
- The reason in focusing on Haemonchus contortus is well-explained, but the introduction could provide more context on the economic and animal health importance of this parasite in the specific geographical region of the study (North-Eastern Italy). is there specific regional challenges or characteristics that make this research particularly important?
Materials and Methods
- The description of the primer and probe design is somewhat limited. Please give greater detail on the sequence alignment process, the criteria used for primer selection (e.g., melting temperatures, GC content), and any steps taken to avoid non-specific amplification.
- In the specificity testing part, it is reported that adult parasites were identified "according to existing morphological and morphometric keys." Please provide more references for these keys.
- DNA extraction protocols are describe, but it would be helpful to include information in any quality control measures used to assess the extracted DNA (e.g., spectrophotometry).
- For the FECRT trials, the rationale for using visual assessment for weight estimation should be further justified. While the authors mention validation during the training phase, more detail on the accuracy of this method would be beneficial.
- The pooling of fecal samples is described, but the potential impact of pooling on the sensitivity of the real-time PCR assay should be addressed. Could pooling lead to a loss of information on individual animal infection levels, and how might this affect the interpretation of FECRT results?
Results
- Table 2 presents the results of the molecular testing. It would be helpful to include standard deviations or other measures of variability for the "GEN Estimated quantity" and "HAEM Estimated quantity" to provide a sense of the precision of the measurements.
- In the FECRT results, the authors mention that "the treatment was effective in 2 out of 5 sheep trials and in 1 out of 5 goat trials." Please clarify what criteria were used to define "effective." This should directly relate to the efficacy classification (S, R, LR, INC) described in the Methods.
- The discussion of Haemonchus sp. response to treatment in trials G10, S7, S8, and S11 suggests potential resistance. However, the interpretation of FECRT data can be complex. Please provide a more in-depth discussion of the criteria used to differentiate between treatment failure and anthelmintic resistance in this context, considering both FECR values and confidence intervals.
Discussion
- This section effect highlights the advantage of the used real-time PCR techniques. but, it could be improve by a more critical appraisal of the method's limitats. as example, are there any potential biases or sources of error that could affect accuracy Haemonchus sp. quantification?
- implications of widespread trt ineffectiveness are reported, however, authors could improve on the potential consequences of this for animal’s health, wellbeing, and the economic sustainability of sheep and goat farming in the investigated region.
- Discussion conclude with a call for further projects and training. It would be valuable to suggest specific key areas for future research, such as:
- Validation of the assay with a larger sample size and across different geographical regions.
- Comparison of real-time PCR result with other diagnostic methods/techniques.
- Longitudinal research monitoring changes in Haemonchus sp. populations and anthelmintic resistance over-time.
- I recommend discussing the dissemination of your results on Haemonchus contortus diagnosis and anthelmintic resistance in sheep and goat farms via social media to counteract misinformation in animal health.
- Given the prevalence of misinformation in animal health, emphasize sharing your Haemonchus contortus and FECRT findings via social media. Discussing the potential impact on public understanding and awareness. Considering leveraging platforms to disseminate accurate information on parasite diagnosis and anthelmintic resistance beyond the scientific community.
- reporting transparent communication and public engagement through accessible channels will enhance research credibility and foster a better informed society/person. By embracing modern communication tools, you can combat mis-information and ensure your great contributions on Haemonchus contortus and anthelmintic resistance have a meaningful impact.
- I propose adding: "In a parallel vein, a study focused on utilizing Instagram illustrates how social media can serve as an effective tool (10.3168/jds.2024-25347). This study underscores the power of social media in conveying complex topics, such as Haemonchus contortus diagnosis and anthelmintic resistance, to a broad audience. Such initiatives complement the role of influencers in animal health communication by providing tangible examples of how digital platforms can foster community engagement and awareness in specialized areas."
Reference:
I suggest improving the reference list by reporting these valuable papers:
https://pubmed.ncbi.nlm.nih.gov/27238009/
https://www.cambridge.org/core/journals/journal-of-helminthology/article/genotyping-of-benzimidazole-resistance-using-tubulin-isotype-1-marker-in-haemonchus-contortus-of-sheep-and-goats-in-parana-southern-brazil/678E4DA0FBFEAE700E0EC64640ED2F2F
https://avmajournals.avma.org/view/journals/javma/233/12/javma.233.12.1913.xml
Minor comments
Introduction
- Consider rephrasing the sentence "Effective diagnostic tools are critical to monitor the infection level and the efficacy of anthelmintics" for improved clarity and flow.
Materials and Methods
- Please check for consistency in the use of abbreviations (e.g., spell out "polymerase chain reaction" at first use).
- Tab 1, ensure that the 5' and 3' designations are consistently applied to all sequences.
- Check for typos and grammatical errors throughout the manuscript, I found several.
I believe that these comments are helpful for the authors in revising their research paper.

Author Response
The paper written by Maurizio et al. presents a research in the development and use of a novel real-time PCR method for the diagnosis and related quantification of Haemonchus sp., a significant parasitic nematode in graze ruminants. The authors highlight the importance of accurate and timely diagnosis of gastrointestinal nematode infections in livestock, emphasizing the limitations of traditional diagnostics tools and the increasing need for more sensitive and specific molecular techniques.
I found the subject interesting of this research to be fascinating and on time, and I read the manuscript with considerable interests. The paper addresses a critical need in veterinary parasitology, and the development of a method to quantify Haemonchus sp. relative to other strongylids is a valuable contribution. The paper aligns well with the scope of the journal, and the potential applications of this methodology in improving our understanding of anthelmintic resistance and parasite management are significant.
anyway, the study has many strongs, I think that the manuscript, in its present form, has several shortcomings that need to be addressed to enhance its clear, rigor, and general impact. My major comments to the authors, listed below, focus on aspects where I believe revisions are needed to support the conclusions of the research and to ensure the work meets the elevate standards of the journal.
Major Comments to the Author
Introduction
The introduction give a well general overview of the problem of GIN infections and the limitations of traditional diagnostic techniques. However, it could benefit from a clearer statement of the specific aims of the study and the novelty of the developed PCR assay. Consider explicitly stating how this assay improves upon existing molecular diagnostic techniques, beyond just the inclusion of a strongyle quantification.
Reply: We improved the strength of the aim statement as per reviewer’s suggestion.
The reason in focusing on Haemonchus contortus is well-explained, but the introduction could provide more context on the economic and animal health importance of this parasite in the specific geographical region of the study (North-Eastern Italy). is there specific regional challenges or characteristics that make this research particularly important?
Reply: This project was initiated through a collaboration with local veterinarians concerned about suspected anthelmintic failures on farms in the area, with detection of heavy infections of H. contortus during post-mortem evaluations. In Northern Italy, sheep farming is predominantly associated with meat production and it is typically transhumant (this practice was included in the UNESCO Intangible Cultural Heritage), involving the movement of the animals from lowlands in the winter to mountains in the summer. In the alpine area, smaller sheep and goat farms serve both environmental maintenance and local production purposes, while in the Po valley, dairy goats are primarily raised in more modern and intensive farms, frequently with in-loco cheese production. The Po Valley provides an ideal environment for a variety of livestock. However, in this context, small ruminant farming remains a minor and understudied sector, as the focus is primarily on more profitable species like poultry, beef and dairy cattle and pigs, associated to renowned productions such as Parmigiano-Reggiano and Grana Padano cheeses and Parma and San Daniele hams. Despite the global concern over GIN infections and AR, there is notable lack of information on these issues in Northern Italy. This is particularly concerning for small ruminants, which are more vulnerable to these problems due to their more extensive farming system. This study is part of a broader project emerged from the specific needs of the territory and expanded to more methodological levels, aiming to improve AR diagnosis while maintaining a practical approach. As stated, data available on the topic are extremely limited, but we now included two references which highlight the key role of H. contortus in the development of AR in the area.
Materials and Methods
The description of the primer and probe design is somewhat limited. Please give greater detail on the sequence alignment process, the criteria used for primer selection (e.g., melting temperatures, GC content), and any steps taken to avoid non-specific amplification.
Reply: More details on the new method are provided in the revised version
In the specificity testing part, it is reported that adult parasites were identified "according to existing morphological and morphometric keys." Please provide more references for these keys.
Reply: We better explicated the reference for the identification keys.
DNA extraction protocols are describe, but it would be helpful to include information in any quality control measures used to assess the extracted DNA (e.g., spectrophotometry).
Reply: This point has been addressed in the revised version
For the FECRT trials, the rationale for using visual assessment for weight estimation should be further justified. While the authors mention validation during the training phase, more detail on the accuracy of this method would be beneficial.
Reply: We clarified that information on the text.
The pooling of fecal samples is described, but the potential impact of pooling on the sensitivity of the real-time PCR assay should be addressed. Could pooling lead to a loss of information on individual animal infection levels, and how might this affect the interpretation of FECRT results?
Reply: We agree with the reviewer that pooling fecal samples inevitably leads to a loss of information regarding individual animal infection levels. However, this approach reflects standard practice in farm settings, where diagnostics are commonly performed on pooled samples. Our method was specifically designed to operate within a realistic diagnostic context, aiming to be applicable and practical for routine field use. Regarding analytical performance, we do not expect pooling to negatively affect the assay's sensitivity or reliability, as the method was optimized and validated to perform effectively under these conditions.
Results
Table 2 presents the results of the molecular testing. It would be helpful to include standard deviations or other measures of variability for the "GEN Estimated quantity" and "HAEM Estimated quantity" to provide a sense of the precision of the measurements.
Reply: We acknowledge the reviewer’s suggestion to include standard deviation; however, only two replicates were available in this case. With such a limited number of samples, the calculation of standard deviation would not be statistically meaningful. The actual values of both replicates in the manuscript, which provides a direct indication of data dispersion.
In the FECRT results, the authors mention that "the treatment was effective in 2 out of 5 sheep trials and in 1 out of 5 goat trials." Please clarify what criteria were used to define "effective." This should directly relate to the efficacy classification (S, R, LR, INC) described in the Methods.
Reply: We don’t understand the reviewer’s comment. Criteria for efficacy classification are clearly described in the Methods, as stated by the reviewer him/her-self, and the subsequent classification of FECRT results aligned with the criteria.
The discussion of Haemonchus sp. response to treatment in trials G10, S7, S8, and S11 suggests potential resistance. However, the interpretation of FECRT data can be complex. Please provide a more in-depth discussion of the criteria used to differentiate between treatment failure and anthelmintic resistance in this context, considering both FECR values and confidence intervals.
Reply: We thank the reviewer for his/her comment as we agree a comment on these results was missing. We expanded the discussion to clarify our interpretation of these results.
Discussion
This section effect highlights the advantage of the used real-time PCR techniques. but, it could be improve by a more critical appraisal of the method's limitats. as example, are there any potential biases or sources of error that could affect accuracy Haemonchus sp. quantification?
Reply: As with any diagnostic technique, the real-time PCR approach we employed has certain limitations. One potential drawback of increased sensitivity is the heightened risk of cross-contamination; however, this can be effectively minimized through proper sample handling and laboratory procedures. Perhaps the most relevant limitation is the inability of molecular methods to distinguish between DNA from living and dead parasites. While this affects the interpretation of the biological impact of infection at the individual level, it does not interfere with parasite detection or quantification. Additionally, genetic variability could, in theory, influence assay performance. Nonetheless, this is considered a minor concern in our context, as the primers and probes target highly conserved genomic regions, and helminths generally exhibit lower genetic variability compared to other pathogens such as bacteria or viruses.
implications of widespread trt ineffectiveness are reported, however, authors could improve on the potential consequences of this for animal’s health, wellbeing, and the economic sustainability of sheep and goat farming in the investigated region.
Reply: We better addressed this aspect in the discussion.
Discussion conclude with a call for further projects and training. It would be valuable to suggest specific key areas for future research, such as:
- Validation of the assay with a larger sample size and across different geographical regions.
- Comparison of real-time PCR result with other diagnostic methods/techniques.
- Longitudinal research monitoring changes in Haemonchus sp. populations and anthelmintic resistance over-time.
Reply: We expanded the comment on future research as per reviewer’s suggestion.
I recommend discussing the dissemination of your results on Haemonchus contortus diagnosis and anthelmintic resistance in sheep and goat farms via social media to counteract misinformation in animal health. Given the prevalence of misinformation in animal health, emphasize sharing your Haemonchus contortus and FECRT findings via social media. Discussing the potential impact on public understanding and awareness. Considering leveraging platforms to disseminate accurate information on parasite diagnosis and anthelmintic resistance beyond the scientific community. Reporting transparent communication and public engagement through accessible channels will enhance research credibility and foster a better informed society/person. By embracing modern communication tools, you can combat misinformation and ensure your great contributions on Haemonchus contortus and anthelmintic resistance have a meaningful impact. I propose adding: "In a parallel vein, a study focused on utilizing Instagram illustrates how social media can serve as an effective tool (10.3168/jds.2024-25347). This study underscores the power of social media in conveying complex topics, such as Haemonchus contortus diagnosis and anthelmintic resistance, to a broad audience. Such initiatives complement the role of influencers in animal health communication by providing tangible examples of how digital platforms can foster community engagement and awareness in specialized areas."
Reference:
I suggest improving the reference list by reporting these valuable papers:
https://pubmed.ncbi.nlm.nih.gov/27238009/
https://www.cambridge.org/core/journals/journal-of-helminthology/article/genotyping-of-benzimidazole-resistance-using-tubulin-isotype-1-marker-in-haemonchus-contortus-of-sheep-and-goats-in-parana-southern-brazil/678E4DA0FBFEAE700E0EC64640ED2F2F
https://avmajournals.avma.org/view/journals/javma/233/12/javma.233.12.1913.xml
Reply: We thank the reviewer for his/her valuable suggestion. To our experience, many goat and sheep farmers make limited use of technology and we believe that in our context the spread of information through this channel, though certainly useful to raise awareness among some farmers and also veterinarians, would still have some major constraints in our area. For this reason, we didn’t feel comfortable in giving such a strong statement on the issue. However, we strongly agree with the reviewer that social media could have a huge impact in the dissemination of scientific results, and we expect this potential contribution to increase in the future, so we added a final sentence about it. We also included some of the recommended papers in the reference list.
Minor comments
Introduction
Consider rephrasing the sentence "Effective diagnostic tools are critical to monitor the infection level and the efficacy of anthelmintics" for improved clarity and flow.
Reply: Done.
Materials and Methods
Please check for consistency in the use of abbreviations (e.g., spell out "polymerase chain reaction" at first use).
Reply: Done.
Tab 1, ensure that the 5' and 3' designations are consistently applied to all sequences.
Reply: Done.
Check for typos and grammatical errors throughout the manuscript, I found several.
Reply: Done.
Reviewer 3 Report
Comments and Suggestions for Authors
The manuscript by Anna Maurizio and colleagues proposed and validated a novel real-time PCR method for diagnosing Haemonchus sp. infection in grazing ruminants and supporting FECRT-based efficacy assessments. While the study is informative and provides methodological innovation, several aspects warrant further attention:
The manuscript sometimes switches between Haemonchus sp. and H. contortus, though it appears that all data pertain to H. contortus. This should be clarified.
While the ethical exemption is justified, formally including an IACUC or similar waiver ID might strengthen transparency.
Trial G9 (D0) and Trial S10 (D14) yielded >100% relative abundance for Haemonchus sp.—a biological impossibility. This result is likely due to variance in Ct interpolation but should be explicitly flagged and discussed as a limitation of the assay’s proportional estimation model.
While LOD and efficiency are reported, the manuscript could benefit from presenting intra- and inter-assay CVs across independent runs and DNA concentrations.
Author Response
The manuscript by Anna Maurizio and colleagues proposed and validated a novel real-time PCR method for diagnosing Haemonchus sp. infection in grazing ruminants and supporting FECRT-based efficacy assessments. While the study is informative and provides methodological innovation, several aspects warrant further attention:
The manuscript sometimes switches between Haemonchus sp. and H. contortus, though it appears that all data pertain to H. contortus. This should be clarified.
Reply 1: the novel PCR HAEM has been developed to target all species of the genus Haemonchus (e.g., H. contortus, H. placei), therefore in the parts of the manuscript referring to the development of the real-time PCR we preferred to refer to Haemonchus sp., while we used the specific H. contortus term when referring to field samples coming from sheep and goats and to FECRT interpretation, since this is the only one present in small ruminants. However, thanks to reviewer comment, we realized that in many points it was kept the term Haemonchus sp. instead of the specific one. This is now corrected in the revised manuscript and a sentence to clarify this approach has been introduced at the beginning of the Discussion section (lines 394-395).
While the ethical exemption is justified, formally including an IACUC or similar waiver ID might strengthen transparency.
Reply 2: we agree with the reviewer, and we will proceed accordingly in the future. At the time of the beginning of the project we did not ask formally for an official waiver, therefore we do not have any ID or any formal document.
Trial G9 (D0) and Trial S10 (D14) yielded >100% relative abundance for Haemonchus sp.—a biological impossibility. This result is likely due to variance in Ct interpolation but should be explicitly flagged and discussed as a limitation of the assay’s proportional estimation model.
Reply 3: we thank the reviewer for highlighting this point, and we now reworded the sentences at lines 324-326, to clarify the correct interpretation of these results. Furthermore, we included a footnote for the Table 2.
While LOD and efficiency are reported, the manuscript could benefit from presenting intra- and inter-assay CVs across independent runs and DNA concentrations.
Reply 4: We apologize for the missing information. The coefficient of variation (CV) values were calculated for each egg dilution, for both assays and across multiple runs. In all cases, the CV was below 5%, supporting the reliability of the assays. The potential effect of the run was also assessed by fitting a generalized linear model (GLM) including dilution, assay target, and run as explanatory variables. Only dilution and the specific assay (Strongylids vs. Haemonchus) showed statistically significant effects, while no significant run effect was observed. These details are now reported both in the M&M and in the Results sections.
Reviewer 4 Report
Comments and Suggestions for Authors
Maurizio et al conducted a study on the application of a PCR to assesss AR of Hc in sheep and goat farms, which sounds an interesting story, but there are some issues need to be addressed.
Title: the developed PCR method can not assess AR, just for the quarlification of eggs to aid FECRT, thus the title needs to be revised accordingly.
Abstract: It is confusing. How did the authors conduct real-time PCR test for AR? Please supply more details for the methods and results, to let readers more understanding.
Keyword: Some keywords were not accurate, e.g., small ruminants should be replaced with sheep and goats, Haemonchus contortus and 18S-rRNA-ITS1-5.8S-ITS2 should be included.
Results:
1. Why not test samples at 7 dpi? As to some drugs, such as benizmidazole, the effective period is relatively short, thus it is better to include samples from this time point.
2. An existing molecular method can be used for comparision of the real-time PCR in the quarlification of eggs.
Discussion: the comparison of detecting eggs for real-time PCR and other reported molecular or traditional tests can be included in this part, to know the advantage and disadvantage of the real-time PCR.
Conclusions: Please streamline this part, use few lines to summary the main finding, significance and future works.
Author Response
Maurizio et al conducted a study on the application of a PCR to assesss AR of Hc in sheep and goat farms, which sounds an interesting story, but there are some issues need to be addressed.
Title: the developed PCR method can not assess AR, just for the quarlification of eggs to aid FECRT, thus the title needs to be revised accordingly.
Reply 1: we understand the reviewer’s point, although the PCR is part of an approach which contributes to determine the relative abundance of Haemonchus contortus in mixed infection, leading to a specific assessment of AR towards this species, when applied to a FECRT. We slightly modified the title to make this aspect clearer.
Abstract: It is confusing. How did the authors conduct real-time PCR test for AR? Please supply more details for the methods and results, to let readers more understanding.
Reply 2: abstract has a limit in the number of words, hampering the insertion of too many details. However, to our opinion, the description of how real-time PCR was used for testing AR is sufficiently clear: at lines 29-31“This study describes the development of a novel real-time PCR method for diagnosing Haemonchus sp. and its relative abundance in mixed infections in grazing ruminants.” (…) and then at lines 34 “It was then applied in Faecal Egg Count Reduction Test (FECRT) trials”.
Keyword: Some keywords were not accurate, e.g., small ruminants should be replaced with sheep and goats, Haemonchus contortus and 18S-rRNA-ITS1-5.8S-ITS2 should be included.
Reply 3: ‘Sheep’, ‘Goats’ and ‘Heamonchus’ were initially included among key words in the original submission and then changed because of a comment from reviewer 1 in the first round of revision. We inserted them again, changing ‘small ruminants’ to ‘sheep’ and ‘goats’.
Results:
- Why not test samples at 7 dpi? As to some drugs, such as benizmidazole, the effective period is relatively short, thus it is better to include samples from this time point.
Reply 4: As stated in lines 184-186 “Fifteen animals were selected for each farm and a faecal sample was collected from them at day 0 (D0) […] and then about 14 days later (D14) depending on the drug used [22].” These latter words refer to the WAAVP guidelines, stating that “A 10–14 day interval remains valid for sheep, horses and cattle treated with the short-acting drugs, with 14 days recommended for ML drugs.”. Accordingly, in our study we remained within the timeframes abovementioned, and when using benzimidazoles the second sampling was scheduled between day 10 and 14 post treatment, depending on the availability of the farmer. We modified the sentence in the methods to clarify our approach.
- An existing molecular method can be used for comparision of the real-time PCR in the quarlification of eggs.
Reply 5: this point was already addressed in previous round and we specified this limitation at lines 493-503.
Discussion: the comparison of detecting eggs for real-time PCR and other reported molecular or traditional tests can be included in this part, to know the advantage and disadvantage of the real-time PCR.
Conclusions: Please streamline this part, use few lines to summary the main finding, significance and future works.
Reply 6: in the first submission the conclusions section was absent, and it was introduced based on the suggestions of reviewers during the first round. Although the lines 493-503, suggesting the comparison with other diagnostic methods, can fit also in the Discussion section, we believe that it helps a more comprehensive conclusive reasoning if kept in this section.
Reviewer 5 Report
Comments and Suggestions for Authors
Gastrointestinal nematode infections, particularly those caused by the blood-sucking Haemonchus spp., are common in goat and sheep farming, but accurate diagnosis remains rare due to the limitations of traditional methods. This study developed a real-time PCR assay to detect Haemonchus spp. and assess parasite abundance in mixed infections, demonstrating good performance in field trials across farms in Northeast Italy. The results revealed widespread anthelmintic resistance, highlighting the urgent need for improved drug management practices.
On the other hand, there are a few parts of the manuscript that are not understood or the argument is not clear. There are several comments are as follows:
1. For the Abstract part, a few sentences (lines 31–35) are too long and affect clarity. Please consider shortening or splitting them for better readability in abstract.
2. Additionally, when mentioning “optimal efficiency and determination coefficients,” it would strengthen the results section to include the actual values.
3. In Table 2, please explain why some data showed higher GEN and HAEM estimated quantities after treatment compared to before treatment.
4. For Result 3, it would significantly enhance the value of this research if comparisons were made between the same drug’s effects in goats versus sheep, and if resistance levels among different drugs were compared.
Author Response
Gastrointestinal nematode infections, particularly those caused by the blood-sucking Haemonchus spp., are common in goat and sheep farming, but accurate diagnosis remains rare due to the limitations of traditional methods. This study developed a real-time PCR assay to detect Haemonchus spp. and assess parasite abundance in mixed infections, demonstrating good performance in field trials across farms in Northeast Italy. The results revealed widespread anthelmintic resistance, highlighting the urgent need for improved drug management practices.
On the other hand, there are a few parts of the manuscript that are not understood or the argument is not clear. There are several comments are as follows:
- For the Abstract part, a few sentences (lines 31–35) are too long and affect clarity. Please consider shortening or splitting them for better readability in abstract.
Reply 1: one sentence was split in two.
- Additionally, when mentioning “optimal efficiency and determination coefficients,” it would strengthen the results section to include the actual values.
Reply 2: The percentage of efficiency and the determination coefficients (R2) are reported in the Results section at lines 281-282.
- In Table 2, please explain why some data showed higher GEN and HAEM estimated quantities after treatment compared to before treatment.
Reply 3: The issue raised by the reviewer manifested only for HAEM (never for GEN) in three samples: G10, S8 and S9. In our opinion, this could be due to a treatment failure towards Haemonchus combined with the natural fluctuations in the egg emission and with the inevitable imprecision of the molecular approach. We modified the table, displaying one decimal for all numbers (even if corresponding to 0), to improve readability.
- For Result 3, it would significantly enhance the value of this research if comparisons were made between the same drug’s effects in goats versus sheep, and if resistance levels among different drugs were compared.
Reply 4: we guess that reviewer is here referring to Table 3. Anyhow, the aim of the study was the development of a novel realtime PCR for the relative quantification of Haemonchus contortus in sheep and goats, to better interpret the results of FECRT. Therefore, the novel method was applied in a realistic situation where veterinary practitioners decide the drug to be used. The study design was NOT an experimental trial aimed at assessing the efficacy of different drugs in the two ruminant species.
Round 2
Reviewer 1 Report
Comments and Suggestions for Authors
Dear Authors,
Thank you for partially addressing my previous recommendations. Indeed, the revisions made have significantly improved the overall quality of the manuscript. However, there are several critical points with which I strongly disagree:
-
iThenticate Similarity Report: The post-revision similarity index stands at 20%, indicating that one-fifth of the manuscript overlaps with previously published works. This raises serious ethical concerns regarding originality, as such a high degree of textual similarity is not acceptable in scientific publishing.
-
Introduction Section: The first paragraph of the Introduction lacks any bibliographic references, while the second paragraph presents 11 references across 11 lines. This uneven distribution of references is not scientifically balanced and compromises the academic integrity of the section.
-
Ethical Concerns: I strongly disagree with the authors' justification that ethical approval was unnecessary for this research. The claim that verbal consent from the owners replaces a formal written and signed informed consent form is not acceptable. Additionally, the absence of approval from an Institutional Animal Care and Use Committee (IACUC)—or its Italian equivalent, the Comitato Etico e per il Benessere degli Animali (CEBA), which is mandatory for any research involving animals—is a serious ethical shortcoming. In Italy, CEBA committees operate within universities, research centers, and institutions that utilize animals for scientific purposes, and they are responsible for evaluating the ethical aspects of research projects, ensuring compliance with both national and international animal welfare regulations. MDPI journals require either a valid ethics committee approval or signed informed consent forms from animal owners. Verbal authorization is not recognized as sufficient under current ethical standards for scientific research.
-
Economic Feasibility – A Critical Point for Approval: The most critical and determining factor for the acceptance of this manuscript remains the lack of economic analysis. I was explicit in my previous review in stating:
“The discussion lacks a crucial element: the financial feasibility of the newly standardized technique. There is little value in developing a highly sensitive and specific test if rural producers cannot afford it. The discussion should include a cost analysis comparing different diagnostic methods, performance times, and positive and negative predictive values.”
The authors’ response does not satisfactorily address this issue. Simply stating that the method is faster is not a sufficient justification. What is required is a thorough economic feasibility analysis. Such an assessment should have been incorporated during the project’s planning phase. Standardizing a new diagnostic technique must include evaluation of its cost-effectiveness compared to existing alternatives. Speed alone does not validate a method if the associated costs are unaffordable for rural producers, who often manage entire herds under constrained economic conditions.
The authors' statement—“we have emphasized throughout the manuscript the reduced processing time associated with this technique compared to coprocultures. However, we have not conducted a detailed cost analysis, as we believe such an evaluation, based solely on data from our laboratory, would not be relevant to the wider audience”—is not scientifically acceptable. Scientific research demands intensive effort, planning, and above all, relevance and applicability. Including such an explanation in the conclusion section does not reflect the scientific rigor, originality, and societal impact that should be expected of a high-quality study.
The absence of this key component significantly limits the manuscript’s practical contribution, especially considering that the target beneficiaries—rural producers—were only verbally consulted, and no formal consent or ethics approval was obtained.
In light of the serious limitations outlined above, I regret to inform you that I recommend the rejection of this manuscript.
Author Response
Comment 1 - iThenticate Similarity Report: The post-revision similarity index stands at 20%, indicating that one-fifth of the manuscript overlaps with previously published works. This raises serious ethical concerns regarding originality, as such a high degree of textual similarity is not acceptable in scientific publishing.
Reply 1: our writing was fully original and the revision should have even increased the level of originality. Some parts of the M&M (e.g., description of molecular test or interpretation of FECRT results) can be very similar to previous published papers, because they are very standardized. We hardly understand how it showed such a high percentage of similarity and how it was different for the original submission.
Comment 2 - Introduction Section: The first paragraph of the Introduction lacks any bibliographic references, while the second paragraph presents 11 references across 11 lines. This uneven distribution of references is not scientifically balanced and compromises the academic integrity of the section.
Reply 2: we introduced two new references in the first paragraph (line 48 and line 54), although it concerns very general aspects. In previous versions, we decided to include all references related to studies developing new molecular assays, which is the specific argument of our study, for a matter of comparison. These references are compressed in few lines, because we avoided to describe the details for each methodology.
Comment 3 - Ethical Concerns: I strongly disagree with the authors' justification that ethical approval was unnecessary for this research. The claim that verbal consent from the owners replaces a formal written and signed informed consent form is not acceptable. Additionally, the absence of approval from an Institutional Animal Care and Use Committee (IACUC)—or its Italian equivalent, the Comitato Etico e per il Benessere degli Animali (CEBA), which is mandatory for any research involving animals—is a serious ethical shortcoming. In Italy, CEBA committees operate within universities, research centers, and institutions that utilize animals for scientific purposes, and they are responsible for evaluating the ethical aspects of research projects, ensuring compliance with both national and international animal welfare regulations. MDPI journals require either a valid ethics committee approval or signed informed consent forms from animal owners. Verbal authorization is not recognized as sufficient under current ethical standards for scientific research.
Reply 3:
Concerning the ethical issue: our Institution has its ethical committee and before starting the study we informally consulted some members of the committee to verify the need for a formal request. In our case, the study is based on samples derived from monitoring and diagnostic activity conducted by private practitioners in the framework of their professional job, and we basically proposed to different veterinarians to support their diagnostic activities through a structured scheme, using “routine diagnostic sampling procedures”. Considering the absence of an experimental intervention, the formal request to the committee was waived. It’s worth of note that part of the results of this study were already published (doi: 10.1186/s12917-024-04347-7) in an international peer-reviewed journal (BMC Veterinary Research) and our position regarding the ethical issue was accepted. Furthermore, similar research (doi: 10.3390/vetsci8050069) was published in ‘Veterinary Sciences’ in 2021 and again our position was accepted.
Concerning the ‘informed consent’ issue: we agree with the Reviewer that a written informed consent was the best option for this kind of study and we actually started our study accordingly, preparing an informed consent template and sending it to veterinarians involved on the study. Unfortunately, the work on the field is not so simple and we couldn’t collect the signed informed consents from all farmers participating in the survey, and therefore we cannot claim for it. However, we stressed the point with private practitioners collaborating with the study and they received detailed instructions on the protocol for samples collection and analysis and for providing the farmers with the correct information on the aim of the study (attached files, in Italian language). We are confident that all farmers received an appropriate information of these aspects and were verbally informed, although we have a written signed document only for part of them.
We are aware that these aspects are receiving more and more attention in recent years, but we believe that our study is in line with actual ethical requirements (both regarding animal welfare and informed consent of participants), although formal documents were not entirely retrieved.
Comment 4 - Economic Feasibility – A Critical Point for Approval: The most critical and determining factor for the acceptance of this manuscript remains the lack of economic analysis. I was explicit in my previous review in stating:
“The discussion lacks a crucial element: the financial feasibility of the newly standardized technique. There is little value in developing a highly sensitive and specific test if rural producers cannot afford it. The discussion should include a cost analysis comparing different diagnostic methods, performance times, and positive and negative predictive values.”
The authors’ response does not satisfactorily address this issue. Simply stating that the method is faster is not a sufficient justification. What is required is a thorough economic feasibility analysis. Such an assessment should have been incorporated during the project’s planning phase. Standardizing a new diagnostic technique must include evaluation of its cost-effectiveness compared to existing alternatives. Speed alone does not validate a method if the associated costs are unaffordable for rural producers, who often manage entire herds under constrained economic conditions.
The authors' statement—“we have emphasized throughout the manuscript the reduced processing time associated with this technique compared to coprocultures. However, we have not conducted a detailed cost analysis, as we believe such an evaluation, based solely on data from our laboratory, would not be relevant to the wider audience”—is not scientifically acceptable. Scientific research demands intensive effort, planning, and above all, relevance and applicability. Including such an explanation in the conclusion section does not reflect the scientific rigor, originality, and societal impact that should be expected of a high-quality study.
Reply 4: we disagree with Reviewer opinion on this point. The aim of our study was to develop and to test in the field a novel real-time PCR assay that allows for the specific relative quantification of H. contortus in faecal samples from small ruminants, as stated in the Introduction. The economic feasibility, although requested in the previous round, does not represent a fundamental part of the study in our opinion, and none of the studies on novel molecular assay (the ones cited in the Introduction) conducted an economic feasibility of their methods. We explained in our reply that we opted for a relatively inexpensive method, to be integrated in our diagnostic service. At present, the only available alternative (the coproculture) costs 30 € at our Laboratory, while the estimated cost of the combined use of the GEN and HAEM real-time PCR is 20 €, therefore it can be considered affordable by farmers. Again, we accept Reviewer suggestion in general terms and for promoting in the future the adoption of this novel test by farmers and veterinarians, but we do not think that this is a mandatory step at this stage and for this manuscript.

Reviewer 2 Report
Comments and Suggestions for Authors
dear authors, the paper improved a lot after the revisions, i have no further comments
Author Response
dear authors, the paper improved a lot after the revisions, i have no further comments
Reply: we are grateful to the Reviewer for the useful comments that helped us in improving our manuscript.
Reviewer 4 Report
Comments and Suggestions for Authors
The authors have addressed most issues in the previous version, I have no major concern, but the resolution of Figure 1 needs to be improved.
Author Response
Comment 1: The authors have addressed most issues in the previous version, I have no major concern, but the resolution of Figure 1 needs to be improved.
Reply 1: figure 1 has been changed to assure an higher definition